# Psychrotolerant *Mesorhizobium* sp. Isolated from Temperate and Cold Desert Regions Solubilizes Potassium and Produces Multiple Plant Growth Promoting Metabolites

**DOI:** 10.3390/molecules26195758

**Published:** 2021-09-23

**Authors:** Zahoor Ahmad Baba, Basharat Hamid, Tahir Ahmad Sheikh, Saad H. Alotaibi, Hesham A. El Enshasy, Mohammad Javed Ansari, Ali Tan Kee Zuan, R. Z. Sayyed

**Affiliations:** 1Division of Basic Science and Humanities, Faculty of Agriculture, Sher-e-Kashmir University of Agricultural Sciences and Technology, Sopore 193201, India; baba.zahoor@gmail.com; 2Division of Agronomy, Faculty of Agriculture, Sher-e-Kashmir University of Agricultural Sciences and Technology of Kashmir, Sopore 193201, India; tahirkmr@gmail.com; 3Department of Chemistry, Turabah University College, Taif University, P. O. Box 11099, Taif 21944, Saudi Arabia; s.alosaimi@tu.edu.sa; 4Institute of Bioproduct Development (IBD), University Teknologi Malayisa (UTM), Skudai 81310, Johor, Malaysia; 5City of Scientific Research and Technology Applications (SRTA), New Burg Al Arab, Alexandria 21934, Egypt; 6Department of Botany, Hindu College, Mahatma Jyotiba Phule Rohilkhand University Bareilly, Moradabad 244001, India; mjavedansari@gmail.com; 7Department of Land Management, Faculty of Agriculture, University Putra Malaysia, (UPM), Serdang 43400, Selangor, Malaysia; 8Department of Microbiology, PSGVP Mandal’s Arts, Science and Commerce College, Shahada 425409, India

**Keywords:** potassium solubilization, *Mesorhizobium* sp., plant growth promoting traits, cellulase, rhizosphere

## Abstract

Soil potassium (K) supplement depends intensively on the application of chemical fertilizers, which have substantial harmful environmental effects. However, some bacteria can act as inoculants by converting unavailable and insoluble K forms into plant-accessible forms. Such bacteria are an eco-friendly approach for enhancing plant K absorption and consequently reducing utilization of chemical fertilization. Therefore, the present research was undertaken to isolate, screen, and characterize the K solubilizing bacteria (KSB) from the rhizosphere soils of northern India. Overall, 110 strains were isolated, but only 13 isolates showed significant K solubilizing ability by forming a halo zone on solid media. They were further screened for K solubilizing activity at 0 °C, 1 °C, 3 °C, 5 °C, 7 °C, 15 °C, and 20 °C for 5, 10, and 20 days. All the bacterial isolates showed mineral K solubilization activity at these different temperatures. However, the content of K solubilization increased with the upsurge in temperature and period of incubation. The isolate KSB (Grz) showed the highest K solubilization index of 462.28% after 48 h of incubation at 20 °C. The maximum of 23.38 µg K/mL broth was solubilized by the isolate KSB (Grz) at 20 °C after 20 days of incubation. Based on morphological, biochemical, and molecular characterization (through the 16S rDNA approach), the isolate KSB (Grz) was identified as *Mesorhizobium* sp. The majority of the strains produced HCN and ammonia. The maximum indole acetic acid (IAA) (31.54 µM/mL) and cellulase (390 µM/mL) were produced by the isolate KSB (Grz). In contrast, the highest protease (525.12 µM/mL) and chitinase (5.20 µM/mL) activities were shown by standard strain *Bacillus mucilaginosus* and KSB (Gmr) isolate, respectively.

## 1. Introduction

Potassium (K) is one of the main macronutrients and the most abundant cation in higher plants [1], with significant absorption that plays a vital role in metabolic activity, activation of ~80 enzymes, starch synthesis, sugar degradation, photosynthesis, disease resistance, etc. [2,3]. There are four K forms found in the soil, namely, soil minerals, non-exchangeable, exchangeable, and water soluble [4]. Soil minerals make up approximately 90 to 98% of soil K, which is tightly bound and predominantly unavailable for plant uptake [5,6]. Approximately 10% of the soil K consists of the interlayer of the non-expanded form of K-bearing minerals, such as illite, mica, and feldspar [7,8]. Among the diverse K forms in soil, the concentration of soluble K is usually deficient (1 to 2%) [9,10]. Plants primarily absorb K from the soil, and its availability is generally governed by the K dynamics and its total content. The release of non-exchangeable K to the exchangeable form arises when exchangeable and solution K levels are diminished by crop removal, erosion or leaching, and runoff [11,12]. Only a small fraction of the K requirements of the plant is attained by direct contact through root interception, and the major fraction (90%) is held in insoluble forms [13]. The majority of the K required by plants must be transported in the soil to the roots [14], and the movement of K ions is a significant element in the supply of K. This transport happens mostly in soil fluid, the liquid phase of the soil, by mass flow (with the water going to the roots of a plant) and diffusion through the concentration gradient created by the absorbent surface, i.e., roots. Near the roots, the soil solution is rapidly exhausted of nutrients due to removal by the plants [15,16]. Among the different K forms, the most available to plants and microbes are the soluble and exchangeable forms [10]. The available fraction of K is also susceptible to fixation through electrostatic attraction between negatively charged clay lattice layers and K^+^ ions, besides expansive forces due to ion hydration [17]. The minerals primarily accountable for the fixation of K ions include montmorillonite, vermiculite, and weathered mica. The amount of K fixation is directly proportional to the charge density of the interlayer surface of the clay lattice. Hydrous mica clays have been found to fix huge quantities of K, which could not be released even with boiling HNO_3_ [8]. 

Growing amounts of high-yielding fertilizer responsive crop varieties and the intensification of agriculture have led to the depletion of available reserves of various nutrients, including K, at an alarming rate. Imbalanced fertilizer application is also a prominent contributing factor to the K deficiency in agricultural soils. These issues have enabled a new research frontier focused on the discovery of an alternative natural, sustainable and safe method for improving K availability by exploring the plant-growth-promoting bacteria (PGPB) that stimulate plant growth through nitrogen (N) fixation, mineral solubilization (phosphorus, potassium and zinc), organic matter degradation, and phytohormone production [18,19]. Among these soil born PGPB, some bacterial candidates have played an influential role in releasing various essential macro and micronutrients from the minerals in large enough quantities to sustain crop growth. Potassium solubilizing bacteria is one such group of mineral solubilizers that have been reported to release K in its available form from K-bearing minerals such as illite, mica, feldspars, etc., by excreting various organic acids that directly dissolve minerals or chelate silicon ions to release K [20,21,22,23,24,25]. An array of such bacteria, such as *Bacillus* sp. [2,14], *Pseudomonas* sp. [25], *Paenibacillus* sp. [26], *Burkholderia* sp. [21], *Streptomyces* sp. [27] have been documented to show K solubilization potential. Solubilization of K by the bacterial isolates is influenced by factors such as soil pH, the solubility index of a mineral, temperature, moisture, and above all, the nativity of the bacteria. It has been observed that a microorganism from a particular agro-ecosystem may not express its full mineral solubilization potential when applied in an environment different from its original place of isolation. The performance of region-specific microorganisms has always been found outstanding [28].

The Kashmir and Ladakh regions of Jammu and Kashmir State are unique in that they experience more diverse climatic conditions than other states in India. The soils of the extreme northern areas of this Himalayan state harbor cold-tolerant mineral solubilizing microbes, which can be applied for improving the nutrient availability for growing crops on an eco-friendly and long-term basis. The microbes that have attained the adaption of surviving and functioning under cold climatic conditions are referred to as cold-tolerant; such microbes isolated from the Arctic and Himalayas were found to exhibit plant growth-promoting (PGP) activity at a temperature range of 4–25 °C [29,30]. Therefore, the present study was undertaken to screen potent cold-tolerant K solubilizing bacteria for application in crop fields in the temperate and cold desert regions.

## 2. Materials and Methods

### 2.1. Location Description and Collection of Soil Sample

The study area from which the K solubilizing bacteria have been isolated includes the Ladakh region and Kashmir valley of Jammu and Kashmir State, India. The Ladakh region, known as the cold desert, is located at altitudes of 2676 to 3500 m, with a mean annual temperature of 8.6 °C, and receives the scant annual rainfall of 102 mm to 318 mm. The soils are by and large sandy in texture with low organic matter content. Soil reaction is generally alkaline. Drass, a subdivision of the Ladakh region, is recognized as being the second coldest place globally, with mean annual winter and summer temperatures of −12 °C and 23 °C, respectively [31]. Kashmir Valley is a temperate region located at an altitude of above 1850 m.

The soil adhered to the roots of crop plants like maize, oats, wheat, barley, potato, brinjal, tomato, apple, pear, walnut, and almond were collected in sterile polythene bags. Samples from pastures, forests, and glacier sites were also collected from the study area between 2015–2016. The upper surface layer was scrapped, and the rhizosphere soil samples up to the depth of 10 cm were collected. All samples were collected in sterilized zip-locked polybags and stockpiled in an ice chest during transport, preserved at 4 °C under refrigeration conditions in the laboratory, and utilized within 12–24 h after collection for further analysis.

### 2.2. Isolation of K Solubilizing Rhizobacteria

K solubilizing bacteria were isolated on Aleksandrov’s medium containing (g/L) 5.0 g/L glucose, 0.005 g/L magnesium sulphate, 0.1 g/L ferric chloride, 2.0 g/L calcium carbonate, 2.0 g/L K mineral (mica), 2.0 g/L calcium phosphate, 20.0 g/L agar, and 1000 mL distilled water, following modified standard methodology [32]. Serial dilution and pour plating technique was used for isolating the K solubilizing bacteria, followed by incubating the plates at 10 ± 0.2 °C in an inverted position for five days. The colonies that formed hollow zones on the mica amended medium were considered positive for mineral K solubilization and were further purified by repeated streaking on Aleksandrov’s medium, then preserved at 4 °C for further studies [2]. The standard strain *Bacillus mucilaginosus* used as a control was obtained from the Department of Soil Science and Agricultural Chemistry, VN Marathwada Agriculture University, Parbhani, Maharashtra, India.

### 2.3. Screening of K Solubilizing Rhizobacteria

#### 2.3.1. Qualitative Screening for K Solubilization Potential

K solubilization potential of rhizobacterial isolates was qualitatively determined by spot inoculating pure bacterial colonies on Aleksandrov’s medium plates containing mica as insoluble K source and incubating for 20 days. The K solubilization potential was assessed by measuring the diameter of hollow zones formed on the medium. The colony diameter was also measured to work out the solubilization efficiency by using the following formula [33].

#### 2.3.2. Quantitative Screening for K Solubilization Potential

To analyze the K solubilization of isolates, a loopful of 24 h old culture of each rhizobacterial cultures were inoculated into 25 mL of sterilized Aleksandrov’s broth in 50 mL flasks and incubated at 0, 1, 3, 5, 7, 15, and 20 °C for 5, 10 and 20 days on an incubating shaker at 100 rpm. Flasks inoculated with *Bacillus mucilaginosus* were used as controls. The incubated bacterial suspension was centrifuged at 8000× *g* after different time intervals for 10 min to separate supernatant from cell mass, and insoluble K. Supernatant 1 mL was collected in a volumetric flask of 50 mL capacity, and volume was made up to 50 mL with dH_2_O and thoroughly mixed. This solution was fed to a flame photometer to assess the K content [34]. Working standards of 0.0004, 0.0009, 0.001, 0.002 and 0.0024 g KCl solution were utilized to prepare the standard curve and were used for estimating the quantity of K solubilized by the bacterial isolates through the following equation
(1)A=mc
where,

*A* = absorbance of the solution 

*m* = slope of curve, and

*c* = concentration of unknown solution 

### 2.4. Production of Plant Growth-Promoting Characteristics

#### 2.4.1. Hydrogen Cyanide

Qualitative production of hydrogen cyanide was assessed by streaking KSB isolates on King’s B medium augmented with 4.4 g/L glycine. Sterilized filter paper soaked in picric acid solution (2.5 g of picric acid; 12.5 g of Na_2_CO_3_, 1 L of dH_2_O) was positioned inside the upper lid of the petri dish. The dishes were sealed adequately with the parafilm and incubated at 20 ± 1 °C for 48 h. Color change from yellow to different brownish shades was treated as a positive result for hydrogen cyanide production. Based on color change, concentration from yellow to light brown or reddish-brown was considered as weak (+), moderate (++), or strong (+++), accordingly [35].

#### 2.4.2. Ammonia Production

All KSB strains were incubated in peptone water (100 mL) for 4–5 days at 20 ± 2 °C for the determination of ammonia production [36]. After incubation, 1 mL of Nessler’s reagent was supplemented in each tube and the appearance of brown to yellow colour indicated the ammonia production.

#### 2.4.3. Indole Acetic Acid (IAA) Production

IAA production was tested according to the modified standard protocol [37]. Fresh cultures of all KSB isolates were inoculated in 5 mL respective broth tubes and incubated at a specified temperature and time. After incubation, these cultures were centrifuged at 10,000× *g* for 5 min. To 1 mL of supernatant, 4 mL IAA reagent (containing 0.5 M FeCl_3,_ 1 mL, concentrated H_2_SO_4_ 30 mL and distilled water 50 mL) was added, and a final volume of 5 mL was produced, followed by incubation at 20 ± 2 °C for 30 min. The development of a pink color was considered to be positive for IAA production. Using a UV spectrophotometer, absorbance at 530 nm was taken to determine the concentration of IAA in all culture tubes with the help a standard curve prepared with different concentrations of analytical grade IAA.

#### 2.4.4. Protease Activity

The overnight grown bacterial cultures in protease production media were centrifuged at 10,000× *g* for 10 min at 4 °C to collect the supernatant of crude extracellular enzyme, followed by determination of protease activity according to the modified protocol [38]. The reaction mixture contained casein (0.6% *w/v*) in 200 µL (0.05 M) Tris–HCl buffer (pH 7.0) and 100 µL enzyme. For 30 min the reaction mixture was incubated at 20 °C followed by the addition of 300 µL of 10% (*w/v*) trichloroacetic acid (TCA) to stop the reaction. Afterwards, the solution was centrifuged at 16,128× *g* and 4 °C for 10 min, and supernatants (500 µL) were taken by pipetting into separate tubes to which Na_2_CO_3_ (0.4 M, 2 mL) and Folin–Cio-Calteu reagent (1 N, 250 µL) were added. The solutions were incubated at 30 °C for 25 min and absorbance was measured at 660 nm. The protease activity was expressed in µg of tyrosine produced by 1 mL of enzyme in 60 min at 20 °C on tyrosine equivalent. One unit of protease activity was defined as the increase of 0.1-unit optic density at a one h incubation period.

#### 2.4.5. Chitinase Production

The chitinase activity of the bacterial isolates was determined using the modified protocol of [39]. The reaction mixture, containing 2.5 mL of 1% colloidal chitin dissolved in 2.5 mL of phosphate buffer solution and 0.5 mL crude enzyme, was taken in flasks and incubated at 20 °C for 1 h. The reaction was stopped by the addition of dinitrosalicylic (DNS) acid, followed by incubation within a boiling water bath for 10 min. The solution was subjected to centrifugation to collect the supernatant, and the amount of sugar released was estimated at 540 nm (UV spectrophotometer). Under standard assay conditions, one unit of chitinase activity (U) was defined as the quantity of enzyme that yields 1 µM of reducing sugar/minute.

#### 2.4.6. Cellulase Activity

Cellulase activity of the bacterial strains was measured by the following method [40]. The reaction mixtures containing 1.8 mL of 0.5% carboxymethyl cellulose (CMC) were added to 50 mM sodium phosphate buffer (pH 7) and 0.2 mL of crude enzyme in screw capped tubes, and were incubated at 20 °C in a water bath for 30 min. Then 3 mL of DNS reagent was added to each tube to terminate the reaction and centrifuged to estimate the amount of sugars liberated by taking absorbance at 540 nm. Cellulase production was assessed using a glucose calibration curve. One unit (U) of enzyme activity is the amount of enzyme needed under standard assay conditions to release 1 μM of glucose per minute.

### 2.5. Morphological and Biochemical Characteristics

The best KSB isolate was examined for colony morphology, cell shape, and gram reaction, and biochemical characterization according to Bergey’s Manual of Determinative Bacteriology. 

### 2.6. Molecular Characterization and Phylogenetic Analysis of KSB

By using the colony-polymerase chain reaction (PCR) method, the molecular characterization of the selected bacterial isolate was carried out [41]. The universal bacterial primers 27F (5′AGAGTTTGATCCTGGCTCAG3′) and 1492R (5′-GGTTACCTTGTTACGACTT-3′) were used for all bacterial isolates [32]. The PCR reaction was carried out in a final volume of 50 μL containing 3 μL of template DNA, 2 μL of each forward and reverse primer, 0.5 μL of Taq DNA polymerase, 2 μL of dNTP mixture, 5 μL of buffer solution (with MgCl_2_), and 37.5 μL of de-ionized water. Amplification was performed as follows: 30 cycles of 95 °C for 3 min, 94 °C for 30 s, 55 °C for 30 s, 72 °C for 50 s, 72 °C for 8 min. PCR products were sequenced at AgriGenome Lab, Pvt. Ltd., C°Chi, Kerala and the unknown sequence was then identified using the maximum aligned 16S rDNA sequences available in the Gen-Bank of NCBI through BLAST. The phylogenetic tree was built by using the neighbor-joining method.

### 2.7. Statistical Analysis

All experiments were performed in five replicates and the results were stated as means for each sample, including the control group (standard group). The SPSS 16.0 statistical package was used for performing all statistical analyses. One-way analysis of variance (ANOVA) and post hoc analysis (Duncan’s Multiple Range Test) were performed to compare mean values of the isolates and the standard (*p* < 0.05).

## 3. Results and Discussion

### 3.1. Isolation of K Solubilizing Rhizobacteria

From the different soil samples collected from the Ladakh region and Kashmir valley, 110 morphologically different bacteria were isolated. Among these 110 bacterial isolates, only 13 isolates showed K solubilizing ability by forming a halo zone on Aleksandrov’s agar media (Figure 1). In a previous study, the isolation of 15 K-solubilizing rhizobacteria (KSR) from soil samples collected from hilly areas depending on their capability to solubilize mica as insoluble K source was reported [2]. Studies have also shown that different species of KSB exist in the soil and play a vital role in the K cycle [42,43]. Potassium solubilizing bacteria (KSB) are involved in the natural potassium cycle, and therefore, the presence of KSB in the soil makes potassium available for uptake by plants [22,44]. However, few studies on KSB isolation from the cold regions are available in the literature, and the isolates of the present study were cold tolerant for their adaptation of surviving and functioning under cold climatic conditions.

### 3.2. Qualitative Screening for K Solubilization Potential

All thirteen strains were evaluated for the potential qualitative K solubilization under in-vitro conditions. All the strains produced clear solubilization halo zones of different diameters on Aleksandrov’s agar medium, ranging from 3.10–5.27 cm after 48 h of incubation at 20 °C (Table 1). The results reveal that out of thirteen, only seven isolates showed significantly (*p* < 0.05) higher solubilization efficiency in comparison to the standard isolate (*Bacillus mucilaginosus*) (169.79%). However, the highest significant (*p* < 0.05) solubilization efficiency was displayed by isolate KSB (Grz) (462.28%). Solubilization of mineral K compounds by the abundant soil microorganisms is a common phenomenon through the secretion of organic acids and enzymes, and the production of capsular polysaccharides [21,43]. Other investigations have reported that out of 12 psychrotrophic PGPR bacterial species screened qualitatively for K solubilization, the isolate with strain no. IARI-AR26 displayed the largest halo zone radius (5.8 mm) [42]. Moreover, 11 rhizospheric potassium solubilizing bacteria were screened, and the best isolate, UPS1C1, had a maximum zone growth ratio of 5.0 [42].

### 3.3. Quantitative Screening for K Solubilization Potential 

The quantitative K solubilization by all 13 KSB isolates was assessed by incubating them at different incubation periods viz. 5, 10, and 20 days, at incubation temperatures of 0 °C,1 °C, 3 °C, 5 °C, 7 °C, 15 °C, and 20 °C as shown in Table 2, Table 3 and Table 4, respectively. 

The results revealed that the standard isolate showed solubilization activity only at 7 °C, 15 °C, and 20 °C, by producing 0.89, 2.29, and 4.31 µg K/mL broth, respectively. All the isolates except KSB (Sls) and KSB (Trs) produced a significantly (*p* < 0.05) higher quantity of K compared to standard at an incubation temperature of 20 °C. However, the maximum quantity (10.19 µg K/mL broth) of K was solubilized by isolate KSB (Grz), which is 136.4 % more than the standard. Mineral K solubilization by isolates even below sub-optimal temperature conditions supports the cold-tolerant nature of these isolates (Table 2).

K solubilization by different bacterial isolates after ten days of incubation time at 0 °C, 1 °C, 3 °C, 5 °C, 7 °C, 15 °C, and 20 °C incubation temperatures, exhibited that the standard (*Bacillus mucilaginosus*) did not solubilize any quantity of K at 0 °C, 1 °C and 3 °C (Table 3). However, at 5 °C, 7 °C, 15 °C and 20 °C, it solubilized 0.03, 1.64, 4.9 and 8.13 µg K/mL broth, respectively. A total of 9/13 isolates, including KSB(Smg), KSB(Drs), KSB(Gmr), KSB(Mng), KSB(Pd), KSB(Kgl), KSB(Grz), KSB(Phg), and KSB(Glm), produced significant (*p* < 0.05) quantities of K from an insoluble source (mica), while the isolates KSBM, KSB(Sls), KSB(Trs) and KSB (Afr) produced significantly (*p* < 0.05) lower K quantities in comparison to standard at 20 °C incubation temperature. However, the maximum quantity (13.33 µg K/mL broth) of K was solubilized by isolate KSB(Grz), 63.9% more than that produced by the standard.

After twenty days of incubation at 0 °C, 1 °C, 3 °C, 5 °C, 7 °C, 15 °C and 20 °C incubation temperatures, the K solubilization pattern by various isolates revealed that the standard (*Bacillus mucilaginosus*) exhibited solubilization activity only at and beyond 5 °C by producing 0.04, 0.97, 10.76 and 15.33 µg K/mL broth at 5 °C, 7 °C, 15 °C, and 20 °C, respectively (Table 4). Out of 13 KSB isolates, only KSB (Smg), KSB (Drs), KSB (Gmr), KSB (Mng), and KSB (Grz) solubilized significantly (*p* < 0.05) higher K content than standard. However, a significant (*p* < 0.05) maximum of 23.38 µg K/mL broth was solubilized by KSB (Grz), which is 52.5% more than the quantity solubilized by the standard isolate 20 °C. All the K solubilizing isolates showed almost stabilized K solubilization at 15 °C and 20 °C after 20 days of incubation that could be attributed to their psychrotrophic nature, while the standard isolate did not express any stabilization at 20 °C after 20 days of incubation. The ability of KSB bacterial strains to show activity at lower temperatures could be attributed to their adaptability to these conditions [45] via several mechanisms such as cytoplasmic membrane fluidity, synthesis of cold shock proteins, and cryoprotectant genes [46]. Other studies have documented the psychrotrophic PGPR with mineral solubilization potential such as the solubilization of potassium, isolated from the Himalayan regions [42,47,48,49]. The solubilization of K-bearing rocks and minerals occurs through the release of various organic acids and enzymes, accompanied by acidolysis, complex lysis exchange reactions that lead to their conversion to a soluble form. The organic and inorganic acids can release the bound form of K from the unavailable and fixed forms, thereby improving the K availability to the plants [50,51,52]. 

### 3.4. Production of Plant Growth-Promoting Substances

All the K solubilizing isolates exhibited PGP activities such as the production of HCN, ammonia, and IAA (Table 5). The isolates KSB(Smg), KSB(Drs), KSB(Pd), KSB(Sls), KSB(Trs), KSB(Grz), and KSB(Glm) produced significantly (*p* < 0.05) higher IAA content, as compared to standard isolate, with KSB(Grz) producing the highest content of 31.54 µM/mL. The protease activity shown by all KSB isolates was non-significant in comparison to the standard. Only one isolate, KSB(Gmr), showed significantly (*p* < 0.05) higher (5.2 µM/mL) chitinase activity, similarly only two isolates, KSB(Grz) and KSB(Afr), showed significantly higher cellulase activities of 390 and 270 µM/mL, respectively. The potential of cold tolerant bacteria to produce cold active enzymes and plant growth promoting substances has been also documented in an earlier study [53]. The capability of microbes to produce ammonia helps the plants to fulfill their nitrogen needs; the ample supply of ammonia to plants diminishes plant colonization by pathogenic microbes. Microorganisms produce ammonia via the hydrolyzation of urea into ammonia and CO_2_ [54], or through the fixation of atmospheric nitrogen into ammonia that inhibits the growth of pathogenic microbes [55]. Several rhizobacterial strains have been found capable of producing many volatile compounds such as hydrogen cyanide (HCN) to control some soil-borne diseases like root rots in various crop plants [56,57,58]. Microorganisms such as *Pseudomonas*, *Bacillus*, *Bradyrhizobium*, *Klebsiella*
*Rhizobium* sp. etc., were found to be involved in the biosynthesis of IAA through the formation of indole-3-pyruvic acid and indole-3-acetic aldehyde [59,60,61,62,63]. Growth improvement through enzyme activity such as proteases, chitinases, cellulases, phosphatases, etc., is another mechanism used by plant growth-promoting (PGP) bacteria via attacking pathogens through the excretion of cell wall degenerating enzymes [56,64,65,66,67]. KSB (Grz) isolate was noteworthy as it showed outstanding performance regarding the solubilization of insoluble K, and can be utilized as a K biofertilizer in colder regions.

### 3.5. Morphological and Biochemical Characteristics of the Isolate 

Based on morphological and biochemical features, the strain KSB-Grz was found to be rod-shaped and Gram-negative bacterium with white creamy, circular, entire margins, convex, smooth, and opaque colonies appearing on Aleksandrov’s agar media within 2–3 days of incubation at 20 °C. Biochemical analysis results showed that KSB-Grz is positive for catalase, motility, and casein hydrolysis, and negative for citrate utilization, indole, methyl red, and the Voges Proskauer test. The isolate was found to belong to the species of *Mesorhizobium* sp.

### 3.6. Molecular Characterization by 16S rDNA Gene Sequencing Approach

16S rDNA sequencing of the KSB-Grz strain presented utmost resemblance to the reference sequences from members of the *Mesorhizobium* sp. within the GenBank database. The phylogenetic tree of the KSB-Grz strain depicts its position within the *Mesorhizobium* genus (Figure 2). Based on these observations, the KSB-Grz strain was designated as *Mesorhizobium* sp. The gene sequences of the isolate were submitted to NCBI under the Accession number MH503776. The other bacterial isolates obtained from the cold deserts of Indian Himalaya and that exhibit multifunctional PGP activities at lower temperatures belong to *Arthrobacter nicotianae*, *Brevundimonas terrae*, *Paenibacillus tylopili* and *Pseudomonas cedrina* [68], *Bacillus* sp. [42].

## 4. Conclusions

The results of the present study showed that out of a total 110 bacterial isolates, only 13 bacterial isolates showed the ability of potassium solubilization at lower temperature conditions (0 °C, 1 °C, 3 °C, 5 °C, 7 °C, 15 °C and 20 °C) under in-vitro conditions. However, the isolate KSB (Grz) was outstanding with respect to having the most significant and highest K solubilization potential, both qualitatively and quantitatively, in comparison to other strains and the standard strain (*Bacillus mucilaginosus*). All the KSB strains also possessed plant growth promoting activities by producing HCN, ammonia, IAA and lytic enzymes such as cellulase, protease and chitinase with the isolate KSB (Grz) producing IAA and cellulase significantly higher than those produced by the standard strain and the other strains, while the protease and chitinase were not significant. The potential isolate KSB (Grz) was identified as *Mesorhizobium* sp. via 16SrDNA sequencing. 

## Figures and Tables

**Figure 1 molecules-26-05758-f001:**
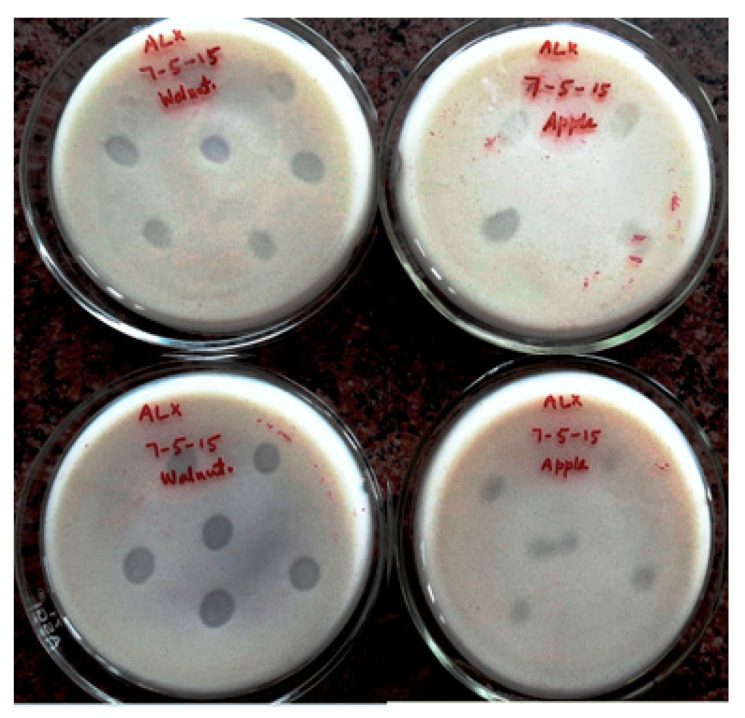
KSB showing solubilization on Aleksandrov’s agar media at 10 °C.

**Figure 2 molecules-26-05758-f002:**
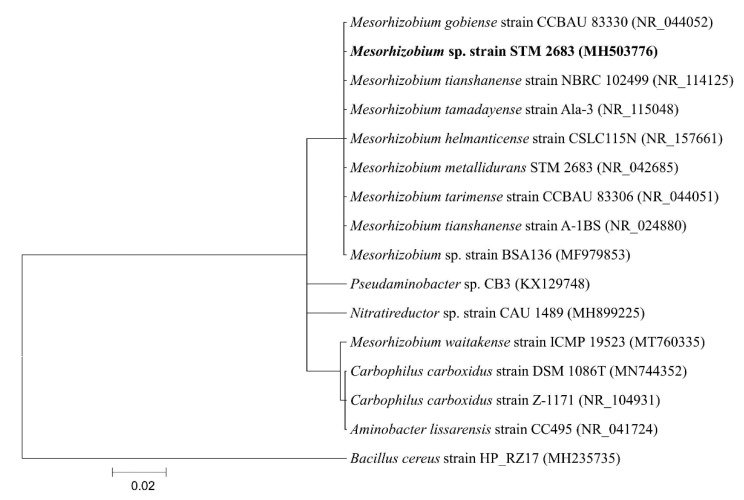
Phylogenetic tree of K-solubilizing KSB (Grz) strain based on 16S rDNA sequences.

**Table 1 molecules-26-05758-t001:** K solubilization efficiency of cold-tolerant bacterial isolates at 20 °C after 48 h.

Isolates	K Solubilization Hallow (cm)	Colony Diameter (cm)	Solubilization Efficiency (%)
*Bacillus mucilaginosus* *	3.26	1.92	169.79 ^h^
KSB(Smg)	3.37	2.82	119.50 ^l^
KSB (Drs)	4.11	1.89	217.46 ^e^
KSBM	3.10	1.98	156.56 ^i^
KSB(Gmr)	3.47	2.61	132.95 ^k^
KSB(Mng)	4.72	1.56	302.56 ^c^
KSB(Pd)	3.25	2.97	109.42 ^m^
KSB(Sls)	3.17	3.12	101.60 ^n^
KSB(Trs)	3.97	2.10	189.04 ^f^
KSB(Kgl)	4.38	1.77	247.45 ^d^
KSB(Grz)	5.27	1.14	462.28 ^a^
KSB(Phg)	3.84	2.18	176.14 ^g^
KSB(Glm)	4.97	1.26	394.44 ^b^
KSB(Afr)	3.61	2.54	142.12 ^j^

* Used as standard. Different letters on the mean values represent significantly (*p* < 0.05) different values according to Duncan’s Multiple Range Test (DMRT).

**Table 2 molecules-26-05758-t002:** Quantitative K solubilization (µg K/mL broth) potential of cold-tolerant bacterial isolates at different temperatures after 5 days of incubation.

Isolates	Incubation Temperature						
	0 °C	1 °C	3 °C	5 °C	7 °C	15 °C	20 °C
*Bacillus mucilaginosus*	0.00 ^i^	0.00 ^n^	0.00 ^h^	0.00 ^l^	0.89 ^n^	2.29 ^i^	4.31 ^l^
KSB(Smg)	1.15 ^c^	1.53 ^f^	1.97 ^bcd^	2.35 ^f^	3.18 ^f^	5.11 ^bcd^	7.41 ^b^
KSB(Drs)	0.89 ^g^	1.618 ^d^	2.01 ^abc^	2.50 ^e^	3.85 ^b^	4.93 ^cde^	5.55 ^i^
KSBM	1.14 ^cd^	1.48 ^g^	1.97 ^bcd^	2.06 ^h^	2.80 ^i^	3.94 ^g^	4.90 ^k^
KSB(Gmr)	1.33 ^b^	1.67 ^c^	2.18 ^ab^	3.10 ^b^	3.55 ^c^	4.70 ^de^	6.51 ^e^
KSB(Mng)	1.09 ^d^	1.15 ^k^	2.00 ^e^	2.70 ^d^	3.21 ^e^	5.45 ^b^	6.55 ^d^
KSB(Pd)	1.10 ^cd^	1.45 ^h^	1.75 ^de^	2.11 ^g^	3.09 ^g^	5.23 ^bc^	7.12 ^c^
KSB(Sls)	0.73 ^h^	0.98 ^l^	1.26 ^f^	1.73 ^j^	2.07 ^l^	3.87 ^g^	4.25 ^m^
KSB(Trs)	0.33 ^j^	0.59 ^m^	0.91 ^g^	1.37 ^k^	1.49 ^m^	2.13 ^i^	4.11 ^n^
KSB(Kgl)	1.30 ^b^	1.57 ^e^	1.83 ^cde^	2.05 ^h^	2.39 ^k^	4.19 ^fg^	5.67 ^h^
KSB(Grz)	1.55 ^a^	1.90 ^a^	2.29 ^a^	3.45 ^a^	4.20 ^a^	7.11 ^a^	10.19 ^a^ (136.4%)
KSB(Phg)	1.00 ^e^	1.26 ^j^	1.58 ^e^	2.03 ^i^	2.91 ^h^	4.55 ^ef^	6.15 ^f^
KSB(Glm)	1.15 ^c^	1.69 ^b^	2.13 ^ab^	2.75 ^c^	3.28 ^d^	5.24 ^cde^	6.13 ^g^
KSB(Afr)	0.95 ^f^	1.36 ^i^	1.72 ^de^	2.033 ^i^	2.46 ^j^	3.89 ^h^	5.31 ^j^

Different letters on the mean values represent significantly (*p* < 0.05) different values according to Duncan’s Multiple Range Test (DMRT).

**Table 3 molecules-26-05758-t003:** Quantitative K solubilization (µg K/mL broth) potential of cold-tolerant bacterial isolates at different temperatures after 10 days of incubation.

Isolates	Incubation Temperature						
	0 °C	1 °C	3 °C	5 °C	7 °C	15 °C	20 °C
*Bacillus mucilaginosus*	0.00 ^n^	0.00 ^n^	0.00 ^n^	0.03 ^n^	1.64 ^n^	4.9 ^n^	8.13 ^j^
KSB(Smg)	1.79 ^b^	3.31 ^b^	4.27 ^a^	3.09 ^a^	5.03 ^d^	8.43 ^b^	11.57 ^b^
KSB(Drs)	1.50 ^h^	2.25 ^g^	2.80 ^g^	3.30 ^g^	5.09 ^c^	7.89 ^f^	9.29 ^h^
KSBM	1.22 ^l^	2.73 ^d^	3.51 ^d^	3.95 ^d^	4.35 ^f^	5.48 ^m^	7.91 ^n^
KSB(Gmr)	1.75 ^d^	3.21 ^c^	4.03 ^c^	4.75 ^c^	5.11 ^b^	8.13 ^c^	11.17 ^c^
KSB(Mng)	1.55 ^f^	2.67 ^e^	2.91 ^f^	3.45 ^f^	4.33 ^g^	7.85 ^g^	10.06 ^f^
KSB(Pd)	1.39 ^j^	1.87 ^k^	2.39 ^i^	3.11 ^i^	3.89 ^j^	6.31 ^i^	8.76 ^i^
KSB(Sls)	1.30 ^k^	1.67 ^l^	2.00 ^l^	2.39 ^l^	3.19 ^l^	5.88 ^j^	8.00 ^l^
KSB(Trs)	1.15 ^m^	1.43 ^m^	1.89 ^m^	2.29 ^m^	3.17 ^m^	5.76 ^l^	7.98 ^m^
KSB(Kgl)	1.77 ^c^	2.53 ^f^	3.19 ^e^	3.90 ^e^	4.49 ^e^	8.11 ^d^	11.00 ^d^
KSB(Grz)	2.15 ^a^	3.67 ^a^	4.10 ^b^	4.79 ^b^	5.29 ^a^	9.20 ^a^	13.33 ^a^ (63.9%)
KSB(Phg)	1.50 ^g^	1.89 ^i^	2.33 ^j^	3.01 ^j^	3.99 ^i^	7.67 ^h^	10.01 ^g^
KSB(Glm)	1.61 ^e^	2.037 ^h^	2.58 ^h^	3.27 ^h^	4.11 ^h^	8.01 ^e^	10.66 ^e^
KSB(Afr)	1.43 ^i^	1.87 ^j^	2.29 ^k^	2.60 ^k^	3.21 ^k^	5.77 ^k^	8.11 ^k^

Different letters on the mean values represent significantly (*p* < 0.05) different values according to Duncan’s Multiple Range Test (DMRT).

**Table 4 molecules-26-05758-t004:** Quantitative K solubilization (µg K/mL broth) potential of cold-tolerant bacterial isolates at different temperatures after 20 days of incubation.

Isolates	Incubation Temperature						
	0 °C	1 °C	3 °C	5 °C	7 °C	15 °C	20 °C
*Bacillus mucilaginosus*	0.00 ^l^	0.00 ^n^	0.00 ^m^	0.04 ^n^	2.97 ^n^	10.76 ^m^	15.33 ^f^
KSB(Smg)	2.11 ^b^	5.21 ^b^	6.89 ^b^	8.41 ^b^	10.58 ^c^	19.73 ^b^	19.72 ^b^
KSB(Drs)	1.90 ^f^	4.91 ^e^	6.10 ^e^	7.01 ^e^	9.13 ^e^	17.30 ^e^	17.31 ^e^
KSBM	1.80 ^h^	4.10 ^f^	5.92 ^f^	6.97 ^f^	8.23 ^f^	12.10 ^i^	14.90 ^g^
KSB(Gmr)	2.05 ^c^	5.04 ^c^	6.67 ^c^	7.25 ^c^	10.27 ^d^	18.34 ^c^	18.34 ^c^
KSB(Mng)	1.95 ^d^	4.97 ^d^	6.43 ^d^	7.11 ^d^	10.89 ^b^	17.83 ^d^	17.84 ^d^
KSB(Pd)	1.84 ^h^	3.70 ^g^	5.05 ^i^	6.08 ^g^	6.78 ^g^	13.09 ^g^	13.10 ^i^
KSB(Sls)	1.67 ^j^	2.78 ^l^	4.09 ^k^	4.90 ^l^	5.55 ^l^	10.97 ^k^	10.97 ^m^
KSB(Trs)	1.50 ^k^	2.47 ^m^	3.85 ^l^	4.44 ^m^	5.13 ^m^	9.50 ^l^	9.80 ^n^
KSB(Kgl)	1.93 ^e^	3.57 ^i^	5.19 ^g^	5.90 ^h^	6.43 ^h^	12.07 ^j^	12.10 ^k^
KSB(Grz)	3.17 ^a^	6.35 ^a^	8.88 ^a^	10.00 ^a^	12.32 ^a^	23.35 ^a^	23.38 ^a^ (52.5%)
KSB(Phg)	1.79 ^i^	3.39 ^k^	5.00 ^j^	5.817 ^j^	6.19 ^j^	12.00 ^j^	12.00 ^l^
KSB(Glm)	1.87 ^g^	3.59 ^h^	5.11 ^h^	5.87 ^i^	6.40 ^i^	13.23 ^f^	13.24 ^h^
KSB(Afr)	1.80 ^h^	3.49 ^j^	5.01 ^i^	5.70 ^k^	6.013 ^k^	12.87 ^h^	12.89 ^j^

Different letters on the mean values represent significantly (*p* < 0.05) different values according to Duncan’s Multiple Range Test (DMRT).

**Table 5 molecules-26-05758-t005:** Production of plant growth-promoting substances and enzymes by K solubilizing bacteria.

Isolates	HCN	Ammonia	IAA (µM/mL)	Protease (µM/mL)	Chitinase (µM/mL)	Cellulase (µM/mL)
*Bacillus mucilaginosus* *	++	++	27.00 ^g^	525.12 ^a^	5.10 ^a^	250.55 ^bc^
KSB(Smg)	+	+	27.22 ^e^	325.74 ^e^	3.70 ^cd^	130 ^e^
KSB(Drs)	+	+	31.00 ^b^	417.24 ^b^	2.90 ^e^	121 ^e^
KSBM			21.37 ^k^	324.19 ^f^	2.70 ^e^	206 ^cd^
KSB(Gmr)	+	+	17.09 ^m^	313.09 ^g^	5.20 ^a^	200 ^e^
KSB(Mng)	-	+	23.11 ^j^	310.23 ^h^	3.80 ^cd^	101 ^e^
KSB(Pd)	+	+	27.18 ^f^	297.12 ^j^	4.70 ^b^	150 ^e^
KSB(Sls)	+	-	29.32 ^d^	281.18 ^k^	3.90 ^c^	185 ^d^
KSB(Trs)	-	+	31.44 ^a^	330.72 ^d^	4.00 ^c^	230 ^bcd^
KSB(Kgl)	+	+	25.71 ^i^	297.88 ^i^	2.90 ^e^	137 ^e^
KSB(Grz)	+	+	31.54 ^a^	397.20 ^c^	3.80 ^cd^	390 ^a^
KSB(Phg)	+	-	26.78 ^h^	267.93 ^m^	3.60 ^d^	229 ^bcd^
KSB(Glm)	+	+	30.31 ^c^	226.19 ^n^	4.00 ^c^	115 ^e^
KSB(Afr)	-	+	19.32 ^l^	278.62 ^l^	3.50 ^d^	270 ^b^

* Used as standard. Different letters on the mean values represent significantly (*p* < 0.05) different values according to Duncan’s Multiple Range Test (DMRT).

## Data Availability

All the data is available in the manuscript.

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
