# Peer review of "Psychrotolerant Mesorhizobium sp. Isolated from Temperate and Cold Desert Regions Solubilizes Potassium and Produces Multiple Plant Growth Promoting Metabolites"

_molecules, 2021, doi:10.3390/molecules26195758_

Round 1
Reviewer 1 Report
Bacteria for converting unavailable and insoluble K forms into plant-accessible forms are very important in crop farming. In this manuscript , 13 bacterial strains were isolated from the cold area in the northern of India and the activities of K solubilizing were also investigated. Some problems were listed as follow:
- The final standard curve formula or chart was not given when making the standard curve of dissolved potassium, and generally five rather than four concentration gradients should be set for the standard curve from the perspective of rigor.
- The study cotent is simple, and no enzyme and gene are investigated concerning in the K-solubilizing.
- The conclusion was not strictly stated, and it should be changed to that part of the isolated strains, namely KSB strain, had potassium soluble ability.
- The conclusion was not strictly stated. It is suggested to specify that the IAA and cellulose produced by KSB (Grz) strain are significantly higher than those produced by standard strain and other isolated strains, while protease and chitinase are not significant.

Author Response
Reviewer 1 Report- Round 1
The authors are thankful to the reviewer for excellent reviewing and valuable suggestions for the improvement of the article. Please find the response for each query and suggestion. We, the authors have incorporated the changes as recommended by the reviewer.
Major Comments
Reviewer 1 Round 1
- The final standard curve formula or chart was not given when making the standard curve of dissolved potassium, and generally five rather than four concentration gradients should be set for the standard curve from the perspective of rigor.
Authors’ response: The final standard curve formula has been introduced (Line No. 157) Five working standards were taken and standard curve was prepared (Line No. 154)
- The study content is simple, and no enzyme and gene are investigated concerning in the K-solubilizing.
Authors’ response: The isolates show the solubilization of insoluble potassium minerals mostly through the production of organic acids and there are no enzymes and their respective genes involved in the process.
- The conclusion was not strictly stated, and it should be changed to that part of the isolated strains, namely KSB strain, had potassium soluble ability.
Authors’ response: The conclusion has been changed to the part of KSB strains showing potassium solubilizing ability (Line No. 374-376).
- The conclusion was not strictly stated. It is suggested to specify that the IAA and cellulose produced by KSB (Grz) strain are significantly higher than those produced by standard strain and other isolated strains, while protease and chitinase are not significant.
Authors’ response: It has been specified that the IAA and cellulase produced by KSB (Grz) strain are significantly higher than those produced by standard strain and other isolated strains, while protease and chitinase are not significant (Line No.377-380).
Reviewer 2 Report
The Manuscript “Psychro-tolerant Desert Rhizobacteria Produces Multiple Plant Beneficial Molecules” has aspects of investigation about screening the potent cold-tolerant K solubilizing bacteria for application in crop fields in the temperate and cold desert regions.
Comments
- Title: the title doesn’t represent completely the manuscript
- Abstract: represent the manuscript
- Additional keywords: are adequate.
- In the Introduction the authors show the proposal and the subject of the manuscript, and the objective is clear.
- Material and Methods is describe satisfactory: -
pag. 4 and others points: the authors could use in unit oC only in 15 oC instead of 0 oC, 1 oC, 3oC, 5oC, 7oC, 15 oC..
alter “ppm” for “g”
pag 4 - “2.3.3. Indole acetic acid (IAA) production” the phrase “To 1mL of supernatant 4 LAA reagent containing FeCl3 (0.5 M) 1 mL, H2SO4 con. 30 mL and distilled water 50 mL was added and a final volume of 5 mL was made up followed by incubation at 20±2 oC for 30 minutes” MUST be review.
pag 5 – “2.6. Molecular characterization and phylogenetic analysis of KSB” the phrase “The PCR reaction was carried out in a final volume of 50μL containing 1 mL of DNA, 2 mL of each forward and reverse primer, 0.5mL of TansTaq DNA polymerase, 2 mL of dNTP mixture, 5 mL of buffer solution (with MgCl2), and 37.5 mL of de-ionized water” MUST be review because the sum of volume used are incorrect. I think that is µL instead of mL
- Result:
In all Table the authors MUST insert the Statistic analysis to means (What are strains that differ of the standard?)
A question principal! I think that the increased (in table 2 – 4 ) is incorrect!!
Example in Table 2 for KSB(Grz) the authors calculated the increase in 57.71% but I think that this value is incorrect (and the others values tables 3 and 4). For me the increase was 136%, because 4.31 is 100% and 10.19 is 236% this way the increase is 136% or {[(10.19./4.31)*100] - 100}
Title Table 5 – unit AIA, Protease, Chitinase, Cellulase (µM//mL)
- The conclusion be according to the results, but I suggest that others interpretation don’t be insert how the last phrase
A question: Why the authors don’t insert the discussion? I suggest that this section may be combined with Results
1) Is the subject of this paper relevant to the Journal? YES
2) Does the paper contain new data or new ideas? YES
3) Is the problem well stated? YES
4) Are the methods suited and adequately described? YES
5) Are the experiments properly planned and executed? YES
6) Are the data sufficient and reliable? YES
7) Are data presented only once? YES
8) Is the discussion consistent with the results? NO (see comments)
9) Is the pertinent literature cited and properly discussed? YES
10) Is the title pertinent and understandable? NO
11) Is the abstract pertinent and understandable? YES
12) Do the keywords accurately reflect the content? YES
Author Response
Reviewer 2 Response
The authors are thankful to the reviewer for excellent reviewing and valuable suggestion for the improvement of the article. Please find the response for each query and suggestion. We, the authors have incorporated the changes as recommended by the reviewer.
Major comments
Reviewer 2 Round 1
Comments
- Title: the title doesn’t represent completely the manuscript
Authors’ response:The title has been changed to completely represent the manuscript.Line No. 1-2
- Abstract: represent the manuscript
Authors’ response:The authors are thankful to the reviewer for the appreciation
- Additional keywords: are adequate.
Authors’ response:The authors are thankful to the reviewer for the appreciation
- In the Introduction the authors show the proposal and the subject of the manuscript, and the objective is clear.
Authors’ response:The authors are thankful to the reviewer for the appreciation
- Material and Methods is describe satisfactory: -
Authors’ response: The authors are thankful to the reviewer for the appreciation
-pag. 4 and others points: the authors could use in unit oC only in 15 oC instead of 0 oC, 1 oC, 3oC, 5oC, 7oC, 15 oC..
Authors’ response: The unit oC has been used only in 20 oC.Line No.149.
alter “ppm” for “g”
Authors’ response: “ppm” has been converted to “g”.Line No. 156-157.
-pag 4 - “2.3.3. Indole acetic acid (IAA) production” the phrase “To 1mL of supernatant 4 LAA reagent containing FeCl3 (0.5 M) 1 mL, H2SO4 con. 30 mL and distilled water 50 mL was added and a final volume of 5 mL was made up followed by incubation at 20±2 oC for 30 minutes” MUST be review.
Authors’ response:The procedure has been reviewed and modified.Line No. 183-184
-pag 5 – “2.6. Molecular characterization and phylogenetic analysis of KSB” the phrase “The PCR reaction was carried out in a final volume of 50μL containing 1 mL of DNA, 2 mL of each forward and reverse primer, 0.5mL of TansTaq DNA polymerase, 2 mL of dNTP mixture, 5 mL of buffer solution (with MgCl2), and 37.5 mL of de-ionized water” MUST be review because the sum of volume used are incorrect. I think that is µL instead of mL
Authors’ response:The procedure has been reviewed and units has been corrected and mL is replaced by µL.Line No. 233-235
- Result:
In all Table the authors MUST insert the Statistic analysis to means (What are strains that differ of the standard?)
Authors’ response:In all Tables (Table1, Table 2, Table 3, Table 4 and Table 5) statistical analysis has been inserted to the means to identify strains that are different from the standard. Line No.279-280, Line No. 327-328, Line No. 332-333, Line No. 336-337, Line No. 366-367
A question principal! I think that the increased (in table 2 – 4 ) is incorrect!!
Example in Table 2 for KSB(Grz) the authors calculated the increase in 57.71% but I think that this value is incorrect (and the others values tables 3 and 4). For me the increase was 136%, because 4.31 is 100% and 10.19 is 236% this way the increase is 136% or {[(10.19./4.31)*100] - 100}
Authors’ response: The calculations for correct value of increase has been done and incorporated in Table 2, Table 3 and Table 4. The values are also put in the Results Section in subsection 3.3 Quantitative screening for K solubilization potential. Line No. 289, Line No. 302, Line No.310.
Title Table 5 – unit AIA, Protease, Chitinase, Cellulase (µM//mL)
Authors’ response:Unit µM//mL is corrected to µM/mL in Table 5.Line No.355-356
- The conclusion be according to the results, but I suggest that others interpretation don’t be insert how the last phrase
Authors’ response:The conclusion is presented according to the results and other interpretation has been removed. Line No.376-382.
A question: Why the authors don’t insert the discussion? I suggest that this section may be combined with Results
Authors’ response: The discussion section has been combined with the results. Line No. 247
Round 2
Reviewer 1 Report
This manuscript is important and acceptable. In addition, some grammatical and detailed changes should be revised in the article. All the modified contents are marked yellow in the attached article, please revised again.
The revised contents are summarized as follows:
- The topic. The original title is not appropriate to the main research content of the article, so it is suggested to modify it.
- KCl label production. The original manuscript is not rigorous enough in making standard melody, so it is suggested to make standard melody and calculate potassium solubility according to the newly given data and formula in the attachment.
- Chart data. Due to the large number of samples, it is suggested to use Duncan’s Multiple Range Test (DMRT) for comparison.
- Conclusion. The summary of the original manuscript is not concise and the key points are not prominent, so it is suggested to revise it according to the contents of the attachment.

Author Response
The authors are thankful to the reviewer for appreciation, excellent reviewing, and some more valuable suggestions for the improvement of the article. Please find the response for each query and suggestion. We, the authors have incorporated the changes as recommended by the reviewer.
Minor comments
Reviewer 1 round 2 Report
Comments and Suggestions for Authors
This manuscript is important and acceptable. In addition, some grammatical and detailed changes should be revised in the article. All the modified contents are marked yellow in the attached article, please revised again.
The revised contents are summarized as follows:
- The topic. The original title is not appropriate to the main research content of the article, so it is suggested to modify it.
Authors’ response: The original title of the manuscript has been modified. Line No. 1-3.
KCl label production. The original manuscript is not rigorous enough in making standard melody, so it is suggested to make a standard melody and calculate potassium solubility according to the newly given data and formula in the attachment.
Authors’ response: A total of 5 standard solutions were prepared during the study and only 4 were mentioned in the manuscript. The potassium solubility of the isolates has been interpreted from all standards and the formula in the attachment, as suggested. Table 2, Table 3, and Table 4.
- Chart data. Due to a large number of samples, it is suggested to use Duncan’s Multiple Range Test (DMRT) for comparison.
Authors’ response: As suggested, Duncan’s Multiple Range Test (DMRT) has been used for the comparison of mean values of the samples. Line No. 287-288, Line No.338-339, Line No. 343-344, Line No. 347-348, Line No. 379-380.
- Conclusion. The summary of the original manuscript is not concise and the key points are not prominent, so it is suggested to revise it according to the contents of the attachment.
Authors’ response: The conclusion is revised according to the content of the attachment. Line No. 403-413.

Reviewer 2 Report
Authors accepted almost all my comments and recommendations, but I suggest that the discussion could be improved!!
Author Response
Reviewer 2 round 2 Report
The authors accepted almost all my comments and recommendations, but I suggest that the discussion could be improved!!
Authors’ response: The authors are thankful to the reviewer for the suggestion for the improvement of the manuscript.
As suggested the discussion has been improved. Line No. 257-259, Lino No. 279-283, 324-327, Line No. 359-360, and Line No. 396-399.
